# Exercise-Boosted Mitochondrial Remodeling in Parkinson’s Disease

**DOI:** 10.3390/biomedicines10123228

**Published:** 2022-12-12

**Authors:** Juan Carlos Magaña, Cláudia M. Deus, Maria Giné-Garriga, Joel Montané, Susana P. Pereira

**Affiliations:** 1Blanquerna Faculty of Psychology, Education and Sport Sciences, Ramon Llull University, 08022 Barcelona, Spain; 2CNC—Center for Neuroscience and Cell Biology, CIBB—Centre for Innovative Biomedicine and Biotechnology, University of Coimbra, 3004-504 Coimbra, Portugal; 3Blanquerna Faculty of Health Sciences, Ramon Llull University, 08025 Barcelona, Spain; 4Laboratory of Metabolism and Exercise (LaMetEx), Research Centre in Physical Activity, Health and Leisure (CIAFEL), Laboratory for Integrative and Translational Research in Population Health (ITR), Faculty of Sports, University of Porto, 4150-564 Porto, Portugal

**Keywords:** Parkinson’s disease, mitochondrial remodeling, exercise, non-pharmacological interventions, secretome, muscle–brain communication

## Abstract

Parkinson’s disease (PD) is a movement disorder characterized by the progressive degeneration of dopaminergic neurons resulting in dopamine deficiency in the *striatum*. Given the estimated escalation in the number of people with PD in the coming decades, interventions aimed at minimizing morbidity and improving quality of life are crucial. Mitochondrial dysfunction and oxidative stress are intrinsic factors related to PD pathogenesis. Accumulating evidence suggests that patients with PD might benefit from various forms of exercise in diverse ways, from general health improvements to disease-specific effects and, potentially, disease-modifying effects. However, the signaling and mechanism connecting skeletal muscle-increased activity and brain remodeling are poorly elucidated. In this review, we describe skeletal muscle–brain crosstalk in PD, with a special focus on mitochondrial effects, proposing mitochondrial dysfunction as a linker in the muscle–brain axis in this neurodegenerative disease and as a promising therapeutic target. Moreover, we outline how exercise secretome can improve mitochondrial health and impact the nervous system to slow down PD progression. Understanding the regulation of the mitochondrial function by exercise in PD may be beneficial in defining interventions to delay the onset of this neurodegenerative disease.

## 1. Introduction

Neurodegenerative disorders encompass a wide range of conditions resulting from a progressive process of degeneration of the function and structure of the central and/or peripheral nervous system [1]. Common neurodegenerative disorders include Alzheimer’s disease (AD), Parkinson’s disease (PD), and amyotrophic lateral sclerosis (ALS). Among them, PD is the second most prevalent neurodegenerative disorder after AD, and the most common disorder that affects motor coordination [1]. In adults, the prevalence of PD ranges from 1 to 2 per 1000 [2]. While PD is rare before 50 years old, PD affects 2% of the population above 65 years old and 4% of the population above 80 years old [3]. Worldwide, people are living longer and the population aging is faster than in the past, so the prevalence of PD is expected to increase at an accelerated pace in the coming years. By 2040, the number of people with PD worldwide is projected to exceed 12 million [4]. Dorsey et. al. labels PD a growing *Pandemic* fueled by an aging population, increased longevity, and the byproducts of industrialization [5]. The annual incidence in high-income countries is 14 per 100,000 people in the total population and 160 per 100,000 people over 65 years old [3]. However, there could be several discrepancies regarding PD incidence data, probably due to methodological differences, particularly differences in case ascertainment and diagnostic criteria used [2]. A measurement of the frequency of PD is lifetime risk, which was estimated to be 2% for 40-year-old men and 1–3% for women in the United States (US) population and taking into account competing risks, such as death from other causes, including cardiovascular diseases or cancer [6]. Although the sporadic form of PD (unknown cause) is the most prevalent, around 10–15% of PD patients exhibit the familial form, resulting from a mutation in one or several specific genes [7]. Currently, at least ten mutated genes have been linked with familial PD, including *α-synuclein* (*PARK1*), *Parkin* (*PARK2*), *ubiquitin C-terminal hydrolase L1* (*UCH-L1* or *PARK5*), *PTEN-induced kinase 1* (*PINK1* or *PARK6*), *DJ-1* (*PARK7*), *leucine-rich repeat kinase 2* (*LRRK2* or *PARK8*), *ATPase 13A2* (*ATP13A2* or *PARK9*), *phospholipase A2 group VI* (*PLA2G6* or *PARK14*), *F-Box protein 7* (*FBXO7* or *PARK15*), and *GRB10 interacting GYF protein 2* (*GIGYF* or *PARK11*) [8,9].

To date, PD remains an incurable disorder and a clear diagnosis method is missing. Pathologically, PD is characterized by degeneration of dopaminergic neurons in the *substantia nigra pars compacta* (*SNpc*) of the midbrain culminating in decreased levels of dopamine in *striatum* [10]. Because of this loss, patients diagnosed with PD suffer with severe motor and non-motor symptoms. Typical motor symptoms include resting tremor or rigidity, slowness in voluntary movements called *bradykinesia*, and difficulty in pronunciation when speaking, while non-motor symptoms could be sleep, cognitive, and autonomic function disorders [11,12].

PD history of disease can be divided in four phases, the prodromal phase and early, mid, and late stages. The prodromal phase may start as early as 5–10 years before diagnosis [13]. Although there is no effective cure for PD, some medication, surgery, and interdisciplinary interventions are able to manage to some extent the PD-associated symptoms. The main families of drugs useful for controlling PD-associated motor symptoms are dopamine agonists such as levodopa (L-DOPA) and monoamine oxidase B (MAO-B) inhibitors [14]. Although the exact mechanism by which dopaminergic cell neurodegeneration occurs is not still clear, some lines of evidence implicate mitochondrial dysfunction as a possible primary cause, even before motor symptoms appear [15,16,17]. Furthermore, mitochondrial dysfunction associated with PD has not only been described in the brain, but also extended to peripheral tissues [18,19,20].

Physical activity (PA) and exercise have a huge potential to enhance well-being [21,22]. Accumulating evidence suggests that patients with PD might benefit from PA in a number of ways, from general improvements in health to disease-specific effects and, potentially, disease-modifying outcomes [23,24]. Accordingly, various forms of PA and exercises have shown a beneficial impact in PD patients, including aerobic exercises, treadmill training, dancing, traditional Chinese exercise, yoga, or resistance training [24]. Herein, we refer to exercise-based interventions as a subset of PA, as described by Caspersen and Powell [25]. According to these studies, exercise-based interventions improve mobility, gait, balance and muscle strength of people living with PD [24]. Exercise is often recommended for pharmacologically treated PD patients, as it alleviates their motor symptoms and cognitive deficit [26].

The American College of Sports Medicine (ACSM) and the Parkinson’s Disease Foundation developed a new infographic to provide safe and effective PA guidance for people with PD, built upon the recently released 11th edition of ACSM’s Guidelines for Exercise Testing and Prescription [27]. The current recommendations include 3 days/week of PA for at least 30 min per session of continuous or intermittent aerobic activity at moderate to vigorous intensity; 2–3 non-consecutive days/week of strength training for at least 30 min per session of 10–15 repetitions for major muscle groups; 2–3 days/week of balance, agility, and multitasking activities possibly integrated into their daily routines; and at least 2–3 days/week of stretching, with daily being most effective [27]. However, exercise is still not routinely implemented as a PD co-adjuvant therapy by individual physicians, maybe because of the incomprehension about the mechanisms mediating the therapeutic effects of exercise, at the level of molecules, cells, and systems [26,28,29]. In healthy subjects, exercise-induced beneficial effects come across as being mediated by enhancement of mitochondrial biogenesis/function and mitophagy [28,29,30]. This is relevant as mitochondrial dysfunction is a key “early” PD-associated phenomenon occurring prior to the onset of motor symptoms [31,32].

Mitochondria are the primary source of ATP synthesis within the cell through the oxidative phosphorylation (OXPHOS) system. Several factors could affect mitochondrial activity, such as oxidative stress, nitric oxide, and substrate availability, compromising mitochondrial ability to properly generate energy, produce metabolic intermediates, buffering Ca^2+^, or even regulate cell death. Within cells, mitochondria are key generators of reactive oxygen species (ROS), whose imbalance underlies oxidative damage in many pathologies. Exercise has been shown to regulate mitochondrial respiration, thus affecting ATP production and overall mitochondrial function. Indeed, both acute and endurance exercise augments state 4 respiration and the respiratory control index (RCR: state 3/state 4) [33,34]. RCR is a very informative measurement to study mitochondrial function, as a change in almost any aspect of OXPHOS will affect RCR, making it also a good indicator of dysfunction. A high RCR implies a high mitochondrial capacity for substrate oxidation and ATP turnover and a low proton leak, with RCR values being substrate- and tissue-dependent [33,34,35]. Increased ROS production during exercise can also have an impact on the oxidative status of the cell. However, depending on the mode, intensity, and duration of exercise, the amount of ROS could determine the type of response from oxidative damage to adaptive signaling responses [36].

This review aims to debate the effects of exercise-based interventions to counteract the observed mitochondrial dysfunction in PD patients. Our secondary aim was to analyze the effects of such interventions in improving patients’ quality of life. This critical discussion may sensitize neurologists of the importance of prescribing non-pharmacological therapies, such as exercise as a complementary intervention or as a therapy, at the time of PD clinical diagnosis by making evident the beneficial effects of exercise in PD patients’ life, and thus slowing PD progression.

## 2. Neuronal and Muscular Alterations Found in Parkinson’s Disease

### 2.1. Muscular Function Impairments in Parkinson’s Disease

As the most common movement disorder, PD is associated with primary and secondary motor symptoms. While the most common primary motor symptoms are *akinesia* (lack of spontaneous voluntary movement), *bradykinesia* (slowness of movement), resting tremor, rigidity, and postural instability, secondary motor symptoms include gait disturbance, micrography, precision grip impairment, and speech problems [37]. Collectively, the main motor symptoms are called “*Parkinsonism*”.

Considering that there are no gold standard laboratory or imaging tests allowing a straightforward diagnosis of PD, commonly, the PD clinical diagnostic is based on the presence of two or more PD-associated motor symptoms, including *bradykinesia* in combination with resting tremor or rigidity [38,39,40]. Moreover, relief of motor symptoms with L-DOPA treatment tends to confirm the PD diagnosis [40]. Early symptoms normally are asymmetrical [41]. However, PD-associated symptoms normally develop slowly over time and it is estimated that at least 60% of dopaminergic neurodegeneration occurs when the first motor symptom appears [42].

The severeness of the disease is measured by the unified Parkinson’s disease rating scale (UPDRS), which is the most common metric for clinical studies. However, a modified version known as the Movement Disorder Society-UPDRS (MDS-UPDRS) is used [43], and the Hoehn and Yahr scale is frequently applied to measure the PD stage of progression [44]. With disease progression, the severity of motor and non-motor symptoms increases [45]; the Hoehn and Yahr scale defines five basic stages of progression and is anchored in the distinction between unilateral (stage I) and bilateral (stage II–V) disease and the development of postural reflex impairment (stage III) as a key turning point in the clinical significance of PD [46]. Most of the motor symptoms of the disease described here result directly from neurodegenerative processes, which will be detailed in the next section.

### 2.2. Neuronal Function Abnormalities: The Multi-Faceted Role of Mitochondria in Parkinson’s Disease

Although the exact mechanism by which dopaminergic cell neurodegeneration occurs is still not clear, there are several mechanisms that contribute to PD pathophysiology. The major mechanisms implicated in dopaminergic neurodegeneration in *SNpc* [47] include accumulation of misfolded protein aggregates, alterations in dopamine metabolism, neuroinflammation, mitochondrial dysfunction, and impaired quality control mechanisms [48].

Taking into account the selective degeneration of dopaminergic neurons of *SNpc*, dopamine itself may be an oxidative stress source inducing alteration in brain homeostasis [49,50]. Dopamine is synthesized from tyrosine by tyrosine hydroxylase (TH) and aromatic L-amino acid decarboxylase [51]. Afterwards, dopamine is stored in synaptic vesicles once uptake by vesicular monoamine transporter 2 (VMAT2) [52]. However, in the presence of excessive cytosolic dopamine outside of the synaptic vesicles, in damaged neurons, dopamine is metabolized by MAO or by auto-oxidation to cytotoxic ROS. Subsequently, cerebral mitochondrial respiration is inhibited and the permeability transition pore opening induced, promoting cell death and neuronal loss [53,54]. Moreover, neuronal loss in PD is also associated with chronic inflammation, which is controlled primarily by microglia, the major resident immune cells in the brain, and by astrocytes and oligodendrocytes [55]. It was demonstrated that microglia activation occurs in *SNpc* and olfactory bulb in both sporadic and familial PD patients [56]. In response to detrimental environmental factors or endogenous proteins, microglia can shift to a more activated state and release ROS, leading to neurotoxicity [56].

Overproduction of ROS can cause mitochondrial damage by various mechanisms, including mutations in mtDNA, impaired mitochondrial oxidative phosphorylation, altering mitochondrial permeability, disrupting Ca^2+^ homeostasis, and so on. Aging is a risk factor for developing sporadic PD, which is associated with a decline in mitochondrial function, particularly increasing mtDNA mutations and oxidative stress, and decreasing respiratory chain activity [57]. The mitochondrial respiratory chain in neurons from idiopathic PD patients presents defects in complex I and II, as well as several deletions in mtDNA related to a decreased mtDNA copy number in *substantia nigra* [58]. It is thus clear that mitochondrial dysfunction based on mtDNA and respiratory chain abnormalities contributes to PD pathogenesis by lowering the threshold for susceptibility to other genetic and environmental insults.

Several genes have been directly associated with mitochondrial dysfunction [8,9]. However, new roles of mitochondrial biology regulation identified novel genes causing mitochondrial dysfunction, including *VPS35* and *CHCHD2* [57]. Further, mutant VPS35 triggers mitochondrial fragmentation, leading to impaired mitochondrial complex I assembly and activity promoting neurodegeneration [59]. Besides, α-synuclein has a non-canonical mitochondrial targeting sequence and was also localized in mitochondrial membranes influencing mitochondrial structure and function [60]. Moreover, increased levels of wild-type α-synuclein induced mitochondrial fragmentation and exacerbated production of ROS in both in vitro and in vivo models [61]. Additionally, in human dopaminergic neurons, it was shown that mutated α-synuclein leads to decreased mitochondrial biogenesis through downregulation of peroxisome proliferator activated receptor gamma coactivator 1 alpha (PGC-1α) [57]. Similarly, several models overexpressing wild-type mutated LRRK2 presented defects in mitochondrial dynamics, increased ROS production, and higher vulnerability to mitochondrial toxins [62]. In addition, it was demonstrated that loss of ATPase Na*^+^*/K^+^ transporting subunit alpha 2 (ATP12A2) impairs glycolysis, impacting cellular bioenergetics and aggravating mitochondrial dysfunction. Furthermore, Zn^2+^ homeostasis is also deregulated by the loss of ATP12A2 through impairing vesicular sequestration, promoting mitochondrial and lysosomal dysfunction, contributing to defective mitophagy [59,63,64]. Mitochondria are unquestionably implicated in PD, with parkin-deficient models displaying several defects in mitochondrial morphology and function [62,65] and defective PINK1 impacting mitochondrial function, especially degradation, trafficking and inducing altered mitochondrial morphology [57,63].

A pivotal link between mitochondrial physiology and the major system of protein degradation, the ubiquitin-proteasome system (UPS), was described, with mitochondrial biogenesis being dependent on the proteasome “*surveillance*” [64,66,67]. Proteasomal activity decline occurs with aging, presumably leading to harmful consequences such as the deterioration of mitochondrial functionality and the reduction of cellular proteostasis. In fact, several pieces of evidence showed that deregulation of the proteolytic system contributed to the development of both forms of PD [68,69]. Effectively, a hallmark of PD is the presence of aggregated α-synuclein, ubiquitin, neurofilaments, and molecular chaperones, present as an intraneuronal inclusions called Lewy Bodies (LBs), showing that aberrant protein homeostasis results in toxic accumulation of intracellular proteins, leading to neuronal loss [70,71]. In healthy conditions, wild-type α-synuclein is available in its native conformation as soluble monomers, which mediates α-synuclein physiological function in presynaptic terminals [72]. Moreover, the oligomers, proto-fibrils, and fibrils of α-synuclein or other misfolded proteins can form pores in a protein–membrane model inducing neuronal death through oxidative stress, energy failure, excitotoxicity, and neuroinflammation [73], providing a thorough overview of how a dysfunctional UPS can condition cellular fate and contribute to PD development. However, the regulation of mitochondrial function by UPS and the physiopathological implications in PD remain an emerging question with numerous aspects requiring clarification to potentiate PD amelioration by targeting mitochondrial dysfunction.

## 3. Ameliorating Mitochondrial Dysfunction in Parkinson’s Disease through Exercise

### 3.1. Types of Exercise Applied to Parkinson’s Disease Patients

The characteristic motor and nonmotor impairments in PD patients might motivate the individuals to adopt a sedentary lifestyle, reflecting a deliberate compensatory strategy to prevent further complications, observed, for example, in patients with severe postural instability who try to avoid falls by staying indoors. Indeed, fear of falling is common in patients with PD, and might cause a reduction in their outdoor physical activities [74]. Nevertheless, this precautionary physical inactivity can have deleterious assets in several clinical domains of PD. Accumulating evidence suggests that patients with PD might benefit from PA and exercise in a number of ways, from general improvements in health to disease-specific effects, finally improving their quality of life [75]. For this reason, the ACSM has recently published several guidelines to implement exercise in PD patients in order to create a competency framework for professionals to ensure that people with PD are receiving appropriate, safe, and effective instruction and programs [27].

The different types of exercise programs that bring forward an integral impact on the physical and mental wellbeing of patients with PD include muscular relaxation and activation exercises [76], treadmill gait training [77,78,79,80], body-weight-supported treadmill training [81,82,83], and robot-assisted gait training [84,85]; virtual reality [86,87,88]; aerobic exercise training mainly stationary bicycle [89,90,91,92,93,94,95,96]; balance training [97,98]; exercise + games = exergames [99,100]; high-intensity eccentric resistance training [101]; progressive resistance training [102]; and qigong, tai chi, tango, and yoga [103,104]. The impact of most of these interventions also translates into improved quality of life [81,84,92,98,101,103,105,106,107].

Scientific evidence increasingly reinforces exercise-based interventions as a benchmark for slowing down the neurodegenerative processes associated with PD [105,106,108]. Modest tendencies towards the decrease in oxidative stress and the increase in antioxidant capacity in PD patients have been shown after a resistance training program [107]. Two biomarkers of oxidative stress (malondialdehyde (MDA) and hydrogen peroxide (H_2_O_2_)) were reduced after exercise training (15% and 16%, respectively), highlighting the capacity of the structured physical exercise to improve the oxidative state in people with PD [107].

Batouli and Saba concluded that a large network of brain areas, equivalent to 82% of the total volume of gray matter, was modifiable by PA [109]. The hippocampus was the brain region most affected by exercise [109]. Furthermore, exercise influences neurogenesis in the hippocampal dentate gyrus [110]. According to the prion hypothesis, α-synuclein spreads through neuronal connections and glia [111]. This propagation in the fourth stage of neuropathological course related to PD [112] reaches the *hippocampus* [113], producing changes in the proteins associated with synaptic structures and altering neuronal communication and function [114]. Owing to compensation and plasticity effects, during the progression of PD, training in daily living activities has been shown to restore neural circuits for movement, allowing intact nervous systems [115]. Where they can be key, there exist the cortico-ponto-cerebellar-thalamic-cortical circuits and/or cerebellum-thalamus-ganglia circuits [115], especially in the prodromal period in which non-motor manifestations are described [114,116], because, in stage III, motor symptoms appear as a result of damage in the *SNpc* [112,114].

Some limitations exist to specifying which symptoms respond to exercise, namely the fact that a large number of clinical trials have exclusion criteria in their design that do not allow further study of some of the symptoms and associated comorbidities. In this sense, subjects with orthostatic hypotension, dementia, and comorbidities such as stroke and degenerative osteoarthritis are normally excluded [107]. According to the ACSM, apathy, depression, and fatigue are some of the non-motor symptoms in PD, which can hinder the patient’s ability to participate in physical exercise interventions. Although the benefits of engaging in PA are known, these and other non-motor symptoms are often overlooked and underdetermined when a patient is encouraged to exercise.

### 3.2. Muscle–Brain Crosstalk in Parkinson’s Disease: The Role of Exercise Secretome

For a long time, PD was perceived as a brain disease with neurodegenerative deterioration [117,118]; today, PD is seen as a multiorgan and multisystemic pathology [117,118,119]. Therefore, exercise becomes a strong ally, where accumulating evidence shows that the role of muscular secretory activity, such as adaptation to regular exercise, is crucial [26]. Thus, exercise is responsible for both systemic [120] and neural plasticity [121], which we could call *neural systemic dual plasticity* (Figure 1). This is achieved and developed by the plasticity of the skeletal muscle that has repercussions at the cellular and molecular level [122,123].

The skeletal muscle is an endocrine organ capable of secreting a variety of neurotrophic factors with neuroprotective and beneficial effects [124]. For example, neurological disorders, impaired cognition, dementia, and depression are associated with lower levels of circulating brain-derived neurotrophic factor (BDNF), which can be mitigated by exercise [125]. BDNF release during muscle contraction reaches the brain and binds to tropomyosin receptor kinase B (TrkB), inducing the phosphorylation of several signaling pathways, including phosphoinositide 3-kinase (PI3K)/protein kinase B (AKT)/mammalian target rapamycin (mTOR), PI3K/AKT/mTOR pathway, PI3K/extracellular signal-regulated kinase (ERK)/cAMP responsive element-binding protein (CREB), and Pi3k/ERK/CREB pathway [126,127]. The activation of these signaling pathways leads to the secretion of additional BDNF. Moreover, BDNF promotes the activation of nuclear factor erythroid 2-related factor 2 (Nrf2), which is the master regulator of antioxidant defense systems protecting brain cells from oxidants, inflammatory agents, and electrophiles. Nrf2 system was also implicated in mitochondrial biogenesis and mitochondrial quality control as well with protein homeostasis and cellular redox (Figure 2) [126,127].

Exercise-induced cathepsin B release from skeletal muscle elevates BDNF abundance in the hippocampus, which has been linked to neuroprotection and improved memory function [128]. Many of these molecular adaptations are mediated by PGC-1α (Figure 2) [120].

PGC-1α coordinates the transcription of several biological programs, including mitochondrial function, oxidative metabolism, and Ca^2+^ homeostasis [129], and is also involved in the control of various myokines [130]. PGC-1α increase also leads to the biosynthesis of kynurenine aminotransferases, thus preventing its toxic accumulation in the brain [131]. In addition, it has been shown that exercise enhances neuronal gene expression of fibronectin type III domain-containing protein 5 (FNDC5), whose protein product could stimulate brain-derived neurotrophic factor in the hippocampus (Figure 2) [132]. The endocrine property of muscle cells also includes the release of cytokines (e.g., IL-6) or metabolites (e.g., lactate) [131].

Actually, muscle–brain crosstalk is mediated by myokines and metabolites released by muscle, but the brain also senses exercise indirectly through secretion of adipokines and hepatokines, which can cross the blood–brain barrier [133]. Exercise induces the secretion of the hormone adiponectin derived from adipose tissue, which improves hippocampal neurogenesis and has relevant antidepressant effects [131,134]. In addition, exercise promotes molecules derived from muscle and liver to enter the brain and send signals to receptors located on endothelial, glial, or neuronal cells, thereby triggering the expression of vascular endothelial growth factor (VEGF) and BDNF, key regulators of brain vascularization and plasticity [131], mediating the effects of exercise on the brain. Thus, the identification of exercise-related factors that have a direct or indirect effect on brain function has the potential to highlight new therapeutic targets for neurodegenerative diseases and cognitive enhancers for people of all ages [135].

Information about the specific molecular mechanisms occurring in PD patients’ brains and its modulation by PA is limited, as the study of the human nervous system encounters great difficulty because of its inaccessibility in living patients. Suitable healthy and “only PD” human tissues, uncomplicated by confounding pathologies, are very rarely, if ever, available to investigate. Human brain samples are obtained at autopsy in pathological situations after a variable period without functioning circulation that can markedly influence the amount and state of biomolecules [136]. In addition, cell and animal models do not fully reproduce the pathologies or phenotypes associated with old age [137]. For this reason, the use of peripherally accessible tissues, such as skin cells, has gained interest, making it possible to evaluate mitochondrial bioenergetic defects, to correlate the severity of the symptoms, and to search for biomarkers of the pathogenesis in PD [19,20,138,139].

### 3.3. Exercised Mitochondria, a Cross-Optimization of Metabolic Pathways with Potential to Slow down PD Progression?

Physical activity and exercise affect mitochondrial dynamics and function in all organs in a number of ways that may vary depending on exercise intensity. Acute exercise activates a number of pathways related to PGC-1α, which controls mitochondrial biogenesis through Ca^2+^/calmodulin-dependent protein (CaMK), p38 mitogen-activated protein kinase (MAPK), AMP-activated protein kinase (AMPK), and tumor suppressor protein 53 (p53) signaling. For instance, an increase in cytosolic Ca^2+^ induces PGC-1α, Nrf1, Nrf2, and mitochondrial transcription factor A (TFAM) in L6 myotubes [140]. Further, by increasing PGC-1α promoter activity, the p38-MAPK pathway is also able to induce mitochondrial biogenesis [141]. Additionally, it has been demonstrated that electrical stimulation and exercise result in an increase in AMPK activity [142,143].

The tumor suppressor p53 is also involved in the regulation of mitochondrial biogenesis in cardiomyocytes [138]. Several studies demonstrate that p53 deletion reduces mitochondrial respiration and content decreasing endurance [138,139,144]. In particular, p53 controls mitochondrial respiration by disrupting the equilibrium between glycolytic and oxidative pathways, by translocating to mitochondria and activating TFAM [145,146]. The mechanism appears to be reliant on the amount of training received [147]. In fact, healthy males who performed sprint interval training experienced a considerable increase in their muscle fibers’ maximal mitochondrial respiration, which was correlated with an increase in PGC-1α and p53 levels [147]. Overall, these findings imply that exercise activates different intracellular pathways that favor mitochondria biogenesis, depending on training intensity.

Physical activity and exercise have also been shown to control mitophagy and lysosome biogenesis in cardiac mitochondria [148]. Through promoting mitochondrial turnover, exercise enhances mitochondrial quality and function [148]. For example, acute exercise triggers autophagy in skeletal and cardiac muscle and exercise training induces the selective macroautophagy aimed to remove damaged mitochondria (mitophagy) [148]. As AMPK increases autophagy and given that exercise can induce AMPK, exercise likely induces mitochondrial turnover by activating AMPK-dependent mechanisms [148].

One hallmark of mitochondrial dysfunction is altered mitochondrial morphology a process finely regulated by fusion and fission processes [149]. Exercise affects mitochondrial morphology by activating specific molecular mechanisms. For example, the muscle-specific gene Zmynd17 (MSS51 mitochondrial translational activator) is known to control mitochondrial quality in muscle, especially in fast glycolytic muscles [150,151]. It has also been shown that acute exercise increases mitofusins’ expression in skeletal muscle and stimulates mitochondrial fusion by activating the PGC-1α/ERRα pathway [146]. PGC-1α overexpression in muscle leads to dense mitochondria, increasing endurance [152]. It has been reported that different exercise protocols induce mitochondrial fusion and increased mitofusin 2 (MFN2) [153]. Similarly, following exercise in healthy and moderately active subjects, dynamin-related protein 1 (DRP1) and MFN2 gene expression levels rapidly increased [154,155,156]. In this line, several studies show that acute exercise reduces mitochondrial fission in a β-adrenergic-dependent manner, mainly owing to DRP1 inactivation [152]. Exercise controls fission and fusion processes in skeletal muscle, also affecting Ca^2+^ handling. Indeed, ryanodine receptor 1 fragmentation and the subsequent increase in Ca^2+^ release in the cytosol are acutely induced by high-intensity interval training [157]. Altogether, these findings suggest that exercise turns on particular intracellular pathways to correct abnormalities in mitochondrial dynamics, indicating exercise-induced *intracellular plasticity*.

The PA can also modulate mitochondrial function by altering the mitochondrial respiratory chain plasticity. Electrons produced in diverse metabolic reactions are channeled into the mitochondrial electron transporting chain (ETC) to power the oxidative phosphorylation process. Structurally, ETC encompasses complex I to IV of OXPHOS. ETC plasticity involves plastic changes from freely moving complexes to super-assembled structures, called supercomplexes (SCs), and vice versa [158]. Although the physiology of respiratory SCs formation is not fully understood, there is evidence that they act by minimizing the leakage of electrons, due to an increase in the channeling of the flow of electrons, making mitochondria more efficient and reducing ROS formation [158]. In fact, mitochondrial SCs plasticity has been shown to be influenced by changes in energy demand [159]. A study of physical training in sedentary older subjects showed that exercise affected the stoichiometry of SCs formation in old age [160]. Such evidence reinforces a possible role of PA and exercise in mitochondrial plasticity modulation across different tissues.

Oxidative stress has been implicated in PD [161]. As mentioned, mitochondrial SCs formation can attenuate mitochondrial ROS production. However, mitochondrial ROS production is a natural byproduct of ETC activity, accounting for approximately 90% of cellular ROS. This chronic mitochondrial ROS generation can underlie the oxidative damage in many pathologies. In particular, ROS production during exercise can have an impact on the oxidative status of the cell. Remarkably, although regular PA stimulates health improvements, severe and/or long exercise produces ROS exacerbation, as demonstrated by increased oxidative damage biomarkers in skeletal muscles and blood [36]. Depending on the mode, intensity, and duration of exercise, the amount of ROS could determine the type of response from oxidative damage to adaptive signaling responses [36]. For a synopsis in exercise and oxidative stress research, please consult this work from Powers et al. [162]. Many epidemiological studies support that lifelong exercise decreases the incidence of chronic diseases and reduces all-cause mortality. It is relevant to point out that several of the primary molecular mediators of an exercise stimulus converge on PGC1-α, a master regulator of mitochondrial biogenesis, or are associated with the mitochondrial biology [36].

Although all of these beneficial effects of PA and exercise on enhancing mitochondrial biogenesis and function were known in healthy subjects, it is not clear if these effects can also be obtained in individuals with PD. It is relevant to investigate the effects of different exercise protocols and durations on motor coordination, cognitive aspects, and other (non-)motor symptoms, having mitochondrial performance as a surrogate pattern for the same cohort, and thus be able to verify how an exercise-boosted mitochondrial remodeling impacts the quality of life of patients with PD. By determining mitochondrial biogenesis signaling (PGC-1α, Nrf1, and TFAM), mitochondrial structure (oxidative phosphorylation system protein levels and SCs), mitochondrial network morphology, and mitochondrial respiratory capacity in PD patients, it would be possible to correlate with signs of improvements in motor behavior and quality of life. These studies may enable the characterization of the neuroprotective and motor effects of exercise mediated by mitochondrial modulation specifics of PD patients.

## 4. Conclusions

The present review identifies the effects of exercise-based interventions to counteract the observed mitochondrial dysfunction in PD patients, as well as the effects in improving patients’ quality of life.

Accumulating evidence indicates that PD not only presents with neurodegenerative deterioration but is also considered a multisystem disorder. Although there is no effective cure for PD, some medication, surgery, and interdisciplinary interventions can manage the PD-associated symptoms. Emerging evidence increasingly reinforces exercise as a benchmark for slowing down the neurodegenerative processes associated with PD. PA and exercise have a great impact on cortical activity in PD patients. PA and exercise-based interventions have also shown improvements in health and in disease-specific outcomes, such as improvements in mobility, gait, balance, and muscle strength of people living with PD, as well as enhancements in the mitochondrial function. Therefore, exercise may be responsible for both systemic and neural plasticity. This is achieved and developed by the plasticity of the skeletal muscle and signaling molecules released that have repercussions at the cellular level, namely modulating mitochondrial function. Exercise triggers several changes in the mitochondrial dynamics and function that may be dependent upon exercise intensity.

In terms of quality of life, some types of exercise interventions that have shown an integral impact on the physical and mental well-being of patients with PD include aerobic exercise training, muscular relaxation and activation exercises, treadmill gait training, tai chi, yoga, and so on. Exercise also induces the secretion of the hormone adiponectin derived from adipose tissue, which has potent antidepressant effects.

Overall, further studies are required to identify the best type, intensity, frequency, and duration of exercises to achieve the highest improvements for each disease stage and to promote its prescription among clinicians and health professionals on a regular basis.

## Figures and Tables

**Figure 1 biomedicines-10-03228-f001:**
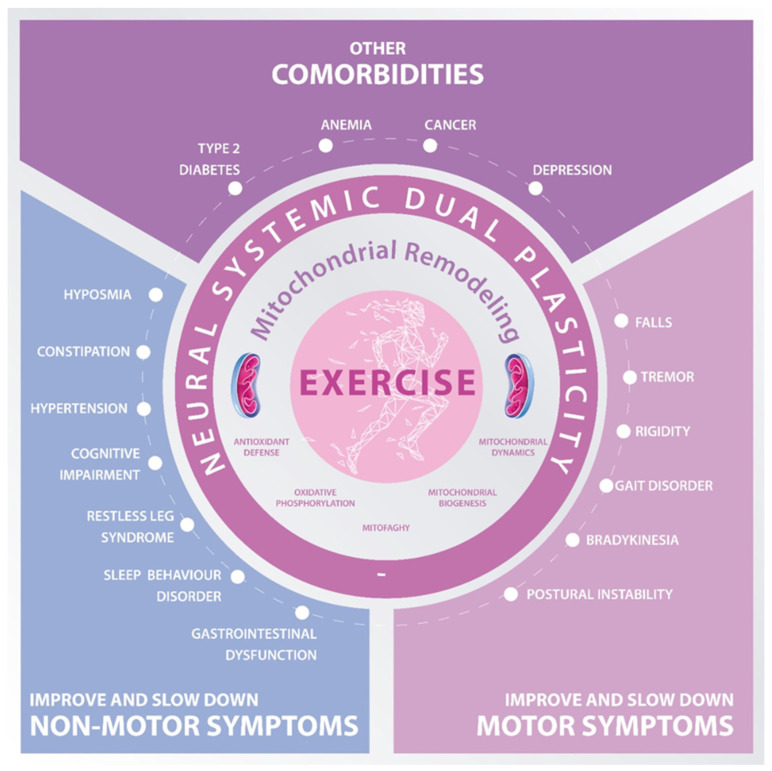
**Quality of life improvements after exercise-boosted mitochondrial remodeling in Parkinson’s Disease.** Exercise promotes a metabolic and mitochondrial remodeling, such as inducing mitophagy, alterations in mitochondrial dynamics, oxidative phosphorylation, and antioxidant defense system. These alterations induce a *neural systemic dual plasticity*, having beneficial effects by preventing other comorbidities, as well as slowing down (non-)motor symptoms, overall improving quality of life in patients with Parkinson’s disease.

**Figure 2 biomedicines-10-03228-f002:**
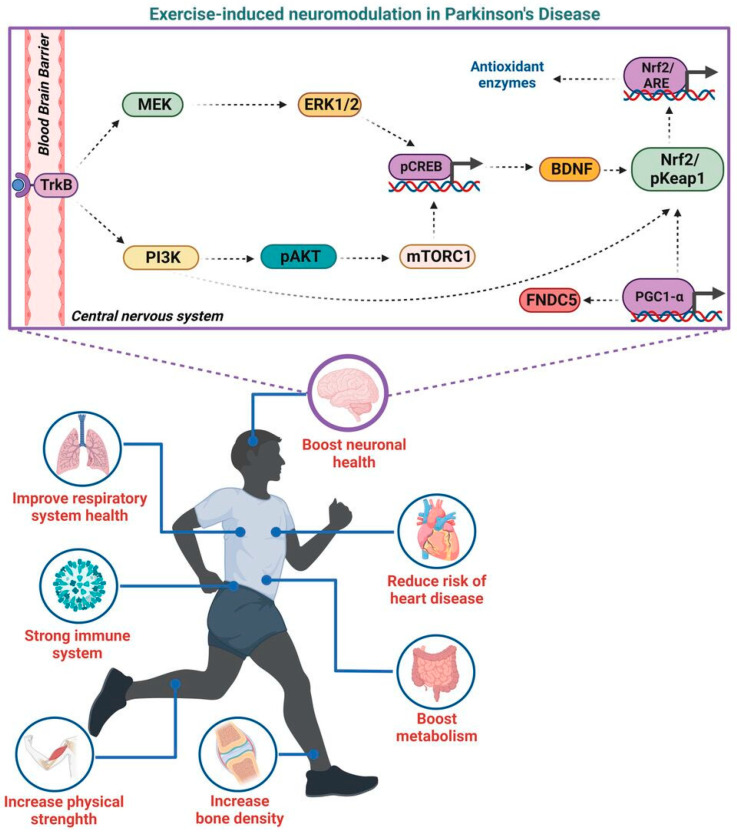
**Effects of exercise in the human body with focus in the brain.** Beside aesthetic issues, being physically active improves respiratory system health, provides a strong immune system, increases muscle strength and bone density, boosts metabolism, reduces the risk of cardiovascular diseases, and boosts mental health. The exact mechanisms by which exercise promotes beneficial effects on motor and non-motor symptoms are not well elucidated. Several myokines secreted in muscles travel in body circulation, reaching the brain and binding to TrkB receptors to trigger different signaling pathways, resulting in the additional secretion of circulating BDNF. BDNF leads to the activation of Nrf2, which regulates the expression of antioxidants molecules. Moreover, expression of PGC-1α is thought to have an important role in the regulation of mitochondrial function and antioxidant defense system activation. Abbreviations: ARE—antioxidant response element; BDNF—brain-derived neurotrophic factor; ERK1/2—extracellular signal-regulated kinase; FNDC5—fibronectin type III domain containing 5; MEK—MAPK/ERK kinase; Nrf2—nuclear factor erythroid 2-related factor 2; pAKT—protein kinase B phosphorylated; pCREB—cAMP responsive element-binding protein phosphorylated; PI3K—phosphoinositide 3-kinase; pKeap1—Kelch-like ECH-associated protein 1 phosphorylated; PGC-1α—peroxisome proliferator-activated receptor-gamma coactivator alpha; TrkB—tropomyosin receptor kinase B. This image was created in BioRender.com (accessed on 17th November 2022).

## Data Availability

Not applicable.

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
