# Peer review of "Exercise-Boosted Mitochondrial Remodeling in Parkinson’s Disease"

_biomedicines, 2022, doi:10.3390/biomedicines10123228_

Round 1
Reviewer 1 Report
The review by Juan Carlos Magaña et al, provides an overview about mitochondrial effects as linker in the muscle-brain axis in Parkinson's disease
The review is clearly written, its original and of interest in its field.
I recommend that the review be accepted with minor revision:
a) The authors mentioned oxidative stress in the abstract section, but did not give due importance in the manuscript. Please describe better the role of the oxidative stress. Incorporating comparisons with other studies would increase the strength of the paper. Please referee doi: 10.1007/s12035-018-1064-2; 10.3233/JPD-130230; 10.3390/biomedicines10051073; 10.1186/s13041-017-0340-9
b) There are some minor grammar issues that should be fixed in order to aid the accessibility of the results to the reader.
Author Response
Barcelona, December 5, 2022
Reference: Comments to the reviewersDear Dr. Cyrus Cheng
We are submitting below the answers to the reviewers’ comments on our review manuscript entitled “Exercise boosted-mitochondrial remodeling in Parkinson’s Disease” by Magaña et al. (Manuscript ID: biomedicines-2059211). We are very thankful for the reviewers’ comments since they helped us improve our manuscript. It is our belief that the manuscript is substantially improved after the suggested edits. We have replied to the reviewers’ concerns and altered the manuscript accordingly. For your review, the alterations made to the manuscript have been done by highlighting in yellow.
Reviewer 1
1) The review by Juan Carlos Magaña et al., provides an overview about mitochondrial effects as linker in the muscle-brain axis in Parkinson's disease
The review is clearly written, its original and of interest in its field. I recommend that the review be accepted with minor revision:
a) The authors mentioned oxidative stress in the abstract section, but did not give due importance in the manuscript. Please describe better the role of oxidative stress. Incorporating comparisons with other studies would increase the strength of the paper. Please referee doi: 10.1007/s12035-018-1064-2; 10.3233/JPD-130230; 10.3390/biomedicines10051073; 10.1186/s13041-017-0340-9
Answer: The authors are grateful for the comment of the reviewer. In the abstract section, authors mentioned oxidative stress as elevated intracellular levels of reactive oxygen species (ROS). In fact, throughout the manuscript authors described the effects of ROS in PD pathogenesis in several sections, including 2.2 (Page 3 and 4 of revised manuscript). Moreover, authors also incorporated comparison with other studies as suggested by the reviewer.
b) There are some minor grammar issues that should be fixed in order to aid the accessibility of the results to the reader.
Answer: As recommended by the reviewer the grammar issues have been revised throughout the manuscript.
We thank all the reviewers for their time, consideration, and comments that helped us to improve our manuscript.
We hope this manuscript is now suitable for publication.
With our best regards,
Sincerely,
Joel Montane
Cláudia M. Deus

Reviewer 2 Report
Strengths:
Description of PD and some examples of molecules and mechanisms involved in muscle exercise and its effects on brain such as BDNF are presented. The language is good although some grammar check is necessary.
Weakness:
The review is rather superficial and a bit chaotic. The introduction is too long as opposed to specific mitochondria-related information.
There is little information on what is going on in the brain actually.
Suggestions:
- It should be indicated which symptoms respond to exercise and which not.
- Exercise guideline should be presented at least briefly - how long, what type of exercise etc.
- The three levels of possible improvement are mentioned but not discussed: general health, improvements to disease-specific effects and disease-modifying effects.
- The long presymptomatic phase of PD and its progressiveness should be also indicated in the Introduction. As well as Lewy bodies.
- Line 86: “Although exercise is often recommended for pharmacologically-treated PD patients, since it alleviates their motor symptoms and cognitive deficit [25], exercise still not routinely implemented as a PD co-adjuvant therapy due to the incomprehension about the mechanisms…” please divide it into shorter sentences. Also, what authors mean exactly by “exercise still not routinely implemented as a PD co-adjuvant therapy” – the official recommendations for PD treatment lists physical activity as adjunct therapy. Do you mean that it is not effectively implemented by the individual physicians?
- Line 95: starting at “Being mitochondria the primary source of ATP synthesis …” please rephrase this sentence as its grammar structure is too complex.
- Line 101 those terms “exercise augments state four respiration and the respiratory control index” need explanation or rephrasing as most readers will not have such specialist knowledge.
- Line 113 – habitual?
- Mitochondria are, not is.
- Paragraph 3 title. Ameliorating mitochondrial dysfunction, not function.
- You should cite directly ACSM’s Guidelines for Exercise Testing and Prescription - not the comment to it. Furthermore there is 11th edition available.
- A clear distinction is needed between which processes are involved in muscles, which in brain and which are general in all organs.
Author Response
Barcelona, December 5, 2022
Reference: Comments to the reviewers
Dear Dr. Cyrus Cheng
We are submitting below the answers to the reviewers’ comments on our review manuscript entitled “Exercise boosted-mitochondrial remodeling in Parkinson’s Disease” by Magaña et al. (Manuscript ID: biomedicines-2059211). We are very thankful for the reviewers’ comments since they helped us improve our manuscript. It is our belief that the manuscript is substantially improved after the suggested edits. We have replied to the reviewers’ concerns and altered the manuscript accordingly. For your review, the alterations made to the manuscript have been done by highlighting in yellow.
Reviewer 2:
1) Strengths: Description of PD and some examples of molecules and mechanisms involved in muscle exercise and its effects on brain such as BDNF are presented. The language is good although some grammar check is necessary.
Answer: As recommended by the reviewer the grammar issues have been revised throughout the manuscript.
2) Weakness: The review is rather superficial and a bit chaotic. The introduction is too long as opposed to specific mitochondria-related information. There is little information on what is going on in the brain actually
Suggestions:
2.1) It should be indicated which symptoms respond to exercise and which not.
Answer: The authors are grateful for the comment of the reviewer. However, there is little information on what is going on in the brain of people suffering with Parkinson’s Disease. Moreover, the symptoms that respond to exercise are conditioned by several other variables, including age, gender, type and frequency of exercise, disease stage, etc. Nonetheless, authors used some studies to emphasize that in section 3.1 (Page 5 and 6 of revised manuscript). Moreover, the following text has been added in section 3.1, page 6:
“Some limitations exist to specifying which symptoms respond to exercise, namely the fact is a large number of clinical trials have exclusion criteria in their design that do not allow further study of some of the symptoms and associated comorbidities. In this sense, subjects with orthostatic hypotension, dementia, and comorbidities such as stroke, degenerative osteoarthritis are normally excluded [112]. According to the ACSM, apathy, depression, and fatigue are some of the non-motor symptoms in PD, which can hinder the patient's ability to participate in physical exercise interventions. Although the benefits of engaging in physical activity are known, these and other non-motor symptoms are often overlooked and undertreated when a patient is encouraged to exercise.”
And the following text has been added in section 3.2, pages 8-9:
“Information about the specific molecular mechanisms occurring in PD patients brains and its modulation by PA is limited, since the study of the human nervous system encounters great difficulty due to its inaccessibility in living patients. Suitable healthy and “only PD” human tissues, uncomplicated by confounding pathologies, are very rarely, if ever, available to investigate. Human brain samples are obtained at autopsy in pathological situations after a variable period without functioning circulation that can markedly influence the amount and state of biomolecules [140]. In addition, cell and animal models do not fully reproduce the pathologies or phenotypes associated with old age [141]. For this reason, the use of peripherally accessible tissues, such as skin cells, has gained interest, making it possible to evaluate mitochondrial bioenergetic defects, to correlate the severity of the symptoms, and also to search for biomarkers of the pathogenesis in PD [19, 20, 142, 143]”.
2.2) Exercise guideline should be presented at least briefly - how long, what type of exercise etc.
Answer: The authors appreciate your comment and we have added this information in the introduction: “The American College of Sports Medicine (ACSM) and the Parkinson’s Foundation developed a new infographic to provide safe and effective PA guidance for people with PD, built upon the recently released 11th edition of ACSM’s Guidelines for Exercise Testing and Prescription [27]. The current recommendations It includes 3 days/week of PA for at least 30 mins per session of continuous or intermittent aerobic activity at moderate to vigorous intensity; 2-3 non-consecutive days/week of strength training for at least 30 mins per session of 10-15 repetitions for major muscle groups; 2-3 days/week of balance, agility and multitasking activities possibly integrated in their daily routines; and at least 2-3 days/week of stretching with daily being most effective [27]” (Page 2 and 3 of revised manuscript).
2.3) The three levels of possible improvement are mentioned but not discussed: general health, improvements to disease-specific effects and disease-modifying effects.
Answer: The authors are grateful for the comment of the reviewer. The objective of this review was not described the three levels of improvement (general health, improvements to disease-specific effects and disease-modifying effects), but describe the skeletal muscle-brain crosstalk in PD, with a special focus in mitochondrial effects, proposing mitochondrial dysfunction as linker in the muscle-brain axis in the neurodegenerative disease and as a promising therapeutic target. Thus, throughout the manuscript authors described “neural systemic dual plasticity”, which is achieved and developed by the plasticity of the skeletal muscle that has repercussions at the cellular and molecular level impacting general health, improvements to disease-specific effects and disease-modifying effects (Page 7 to 9 of revised manuscript).
2.4) The long presymptomatic phase of PD and its progressiveness should be also indicated in the Introduction. As well as Lewy bodies.
Answer: The authors are grateful for the comment of the reviewer. The different phases of Parkinson’s Disease were briefly described in the introduction section (Page 2 of revised manuscript). Considering the Braak model, which proposes six stages of the Lewy pathology in PD progression, authors decided not to mention it since there are several review manuscripts that only focus on explaining that (e.g. PMID: 28243222; 24252164; 29943229).
2.5) Line 86: “Although exercise is often recommended for pharmacologically-treated PD patients, since it alleviates their motor symptoms and cognitive deficit [25], exercise still not routinely implemented as a PD co-adjuvant therapy due to the incomprehension about the mechanisms…” please divide it into shorter sentences. Also, what authors mean exactly by “exercise still not routinely implemented as a PD co-adjuvant therapy” – the official recommendations for PD treatment lists physical activity as adjunct therapy. Do you mean that it is not effectively implemented by the individual physicians?
Answer: As recommended by the reviewer, the authors divided into shorter sentences the sentence of line 86 (Page 3 of revised manuscript). Regarding the use of exercise as a co-adjuvant therapy, the authors know that the official recommendations for PD treatment list physical activity as adjuvant therapy; however, at clinical level it is not effectively implemented by the individual neurologists at the moment of diagnosis of PD.
2.6) Line 95: starting at “Being mitochondria the primary source of ATP synthesis …” please rephrase this sentence as its grammar structure is too complex.
Answer: As recommended by the reviewer the sentence in line 95 was rephrased (Page 4 of revised manuscript).
2.7) Line 101 those terms “exercise augments state four respiration and the respiratory control index” need explanation or rephrasing as most readers will not have such specialist knowledge.
Answer: As recommended by the reviewer the sentence in line 101 was rephrased to improve understanding by readers without that specialist knowledge (Page 3 of revised manuscript)
2.8) Line 113 – habitual?
Answer: The authors are grateful for the comment of the reviewer. The word ‘habitual’ has been replaced by ‘common’ (Page 3 of revised manuscript).
2.9) Mitochondria are, not is.
Answer: The authors are grateful for the comment of the reviewer. The verb “to be” has been changed to plural (Page 3, line 189 of revised manuscript).
3.0) Paragraph 3 title. Ameliorating mitochondrial dysfunction, not function.
Answer: The authors are grateful for the comment of the reviewer. The word “function” has been replaced by “dysfunction”
3.1) You should cite directly ACSM’s Guidelines for Exercise Testing and Prescription - not the comment to it. Furthermore there is 11th edition available.
Answer: The authors appreciate the correction from the reviewer. Authors have updated the reference with the Liguori, G. ACSM's Guidelines for Exercise Testing and Prescription (11th edition).
3.2) A clear distinction is needed between which processes are involved in muscles, which in brain and which are general in all organs.
Answer: The authors are grateful for the comment of the reviewer. As we understand, PD is seen as a multiorgan and multisystemic pathology and it is not yet fully understood which processes affect muscle/brain or all organs in PD patients. Nonetheless, the authors have clarified within section 3.3 the distinction whenever possible and according to the original article (Page 8 to 10 of revised manuscript).
We thank all the reviewers for their time, consideration, and comments that helped us to improve our manuscript.
We hope this manuscript is now suitable for publication.
With our best regards,
Sincerely,
Joel Montane
Cláudia M. Deus

Reviewer 3 Report
This manuscript shows excellent research work. It can be published in this SI of Biomedicines.
Author Response
Barcelona, December 5, 2022
Reference: Comments to the reviewers
Dear reviewer
We are submitting below the answers to the reviewers’ comments on our review manuscript entitled “Exercise boosted-mitochondrial remodeling in Parkinson’s Disease” by Magaña et al. (Manuscript ID: biomedicines-2059211). We are very thankful for the reviewers’ comments since they helped us improve our manuscript. It is our belief that the manuscript is substantially improved after the suggested edits. We have replied to the reviewers’ concerns and altered the manuscript accordingly. For your review, the alterations made to the manuscript have been done by highlighting in yellow.
Reviewer 3:
No comments
We thank all the reviewers for their time, consideration, and comments that helped us to improve our manuscript.
We hope this manuscript is now suitable for publication.
With our best regards,
Sincerely,
Joel Montane
Cláudia M. Deus
